# Learning a Convolutional Bilinear Sparse Code for Natural Videos

**Dimitrios C. Gklezakos**
Paul G. Allen School of CSE
University of Washington
gklezd@cs.washington.edu

**Rajesh P. N. Rao**
Paul G. Allen School of CSE &
Center for Neurotechnology
University of Washington
rao@cs.washington.edu

## Abstract

In contrast to the monolithic deep architectures used in deep learning today for computer vision, the visual cortex processes retinal images via two functionally distinct but interconnected networks: the ventral pathway for processing object-related information and the dorsal pathway for processing motion and transformations [8]. Inspired by this cortical division of labor and properties of the magno- and parvocellular systems [5], we explore an unsupervised approach to feature learning that jointly learns object features and their transformations from natural videos. We propose a new convolutional bilinear sparse coding model that (1) allows independent feature transformations and (2) is capable of processing large images. Our learning procedure leverages smooth motion in natural videos. Our results show that our model can learn groups of features and their transformations directly from natural videos in a completely unsupervised manner. The learned "dynamic filters" exhibit certain equivariance properties, resemble cortical spatiotemporal filters, and capture the statistics of transitions between video frames. Our model can be viewed as one of the first approaches to demonstrate unsupervised learning of primary "capsules" (proposed by Hinton and colleagues for supervised learning) and has strong connections to the Lie group approach to visual perception.

## 1 Motivation

During early development, the brain learns a general-purpose internal representation of objects from unlabeled image sequences. This representation is compositional and leverages the decomposition of objects into parts, sub-parts, and features, along with their relative transformations. In contrast, modern object recognition systems based on deep learning require thousands of labeled examples and typically discard information about transformations (via pooling) in order to achieve invariance. Information about transformations is critical for tasks such as movement planning and spatial reasoning.

Current unsupervised models produce representations that either lack interpretability or hierarchical depth. Variational autoencoders and generative adversarial networks (GANs) typically produce non-interpretable features that do not match the object/parts hierarchy inherent in natural visual scenes. Because they do not explicitly model transformations, these models have difficulty generalizing to the vast range of viewing conditions that objects can appear in. Sparse coding and its variants can learn interpretable features from unlabeled images: these features resemble the localized oriented (Gabor) receptive fields found in the primary visual cortex. However, these models again do not model transformations and have been difficult to generalize to deeper hierarchies due to the combinatorial explosion of possible features.

33rd Conference on Neural Information Processing Systems (NeurIPS 2019), Vancouver, Canada.

We propose a new model for unsupervised learning motivated by the idea that the combinatorial explosion problem can be mitigated by a neural architecture that processes the identity ("what") and the pose ("where") of objects and their parts separately. Such an architecture acknowledges the ventral/dorsal processing dichotomy in the visual cortex: the first is mostly responsible for processing content and identity of objects while the latter is responsible for processing motion and transformations.

We introduce a new bilinear sparse coding model that builds on previous bilinear generative models by (1) allowing each feature to have its own transformation and (2) accommodating large images via transposed convolutions. Furthermore, emulating the slower response times of the parvo pathway compared to the magno pathway, we assume that at short time scales, object identities at each location will remain the same, modeling any fast changes as changes in object transformation values. We demonstrate our model by using short natural video sequences to learn features and their transformations. The resulting collection of "steerable" filters can be viewed as dynamic features resembling the spatiotemporal receptive fields reported in the primary visual cortex. Our model is also one of the first to apply ideas from sparse coding to solve the problem of unsupervised learning of "primary capsules"[1] previously proposed by Hinton and colleagues for supervised learning [4].

## 2 Model

### 2.1 Independent Bilinear Sparse Coding

In bilinear sparse coding [3, 1], an image patch is modeled as a combination of features $B_{ij}$ with two sets of coefficients $r_i$ (object coefficients) and $x_j$ (transformation coefficients) that interact multiplicatively:

$$I \simeq \sum_i \sum_j r_i x_j B_{ij} \tag{1}$$

Let $\sum_j x_j B_{ij} = B_i(\mathbf{x})$ where $\mathbf{x}$ represents the transformation vector consisting of $x_j$'s. Then $I \simeq \sum_i r_i B_i(\mathbf{x})$, which is the standard linear generative model used in sparse coding, PCA, ICA etc. The $r_i$ coefficients correspond to the degree to which each feature exists in the input. The $x_j$ coefficients linearly combine a set of similar features to produce a dynamic "steerable" feature $B_i(\mathbf{x})$. The goal is for these dynamic features to capture an equivariance class centered around an underlying feature $B_i$. As a result, the $r$ coefficients remain invariant regardless of the specific instantiation of the features, the variation being accounted for by $\mathbf{x}$. To learn sparse part-like features of objects, sparsity is enforced on either $r$ or both $r$ and $x$ via some appropriate sparsity penalty.

Typically bilinear sparse coding models are trained using pairs of video frames $I_{t+1}$ and $I_t$, with $r$ fixed and $x$ inferred separately to account for the difference between frames:

$$\Delta I = I_{t+1} - I_t \simeq \sum_i r_i \sum_j \left( x_{t+1,j} - x_{t,j} \right) B_{ij} = \sum_i r_i \sum_j \Delta x_{t,j} B_{ij} \tag{2}$$

There is a strong connection to the Lie group approach to vision [2] where two consecutive frames are modelled as $I_{t+1} = T(\Delta x)I_t$ where $T$ is a transformation operator. The first-order Taylor series approximation of the Lie model [7, 6] is given by: $I_{t+1} = I_t + \sum_j \Delta x_{t,j} \nabla x_j I_t$ which means that $\Delta I = \sum_j \Delta x_{t,j} \nabla x_j I_t$. Suppose $I_t \simeq \sum_i r_i U_i$ where $U_i \in \mathbb{R}^{d \times 1}$ form an underlying feature set. Replacing $\nabla x_j$ with the transformation matrix $G_j \in \mathbb{R}^{d \times d}$, we obtain: $\Delta I \simeq \sum_j \Delta x_{t,j} G_j \sum_i r_i U_i = \sum_i r_i \sum_j \Delta x_{t,j} G_j U_i$. Comparing with Equation 2 above, we see that $B_{ij} = G_j U_i$.

We build on this model by allowing features to have independent pose parameters $x_{ij}$ so that features can transform independently from frame to frame. We also go beyond image patches to modeling large images by using transposed convolutions ($*^T$), resulting in a new bilinear model for images:

$$I \simeq \sum_i r_i \sum_j x_{ij} *^T (G_j U_i) = \sum_i r_i *^T B_i(\mathbf{x}_i) \tag{3}$$

To distinguish our model from past models, we refer to traditional bilinear sparse coding as BSC and our independent bilinear sparse coding model as IBSC.

---

[1]Primary capsules are capsules in the first layer of processing that convert the image into a collection of activations and poses.

## 2.2 Inference

The reconstruction-based loss function for consecutive frames of a video is given by:

$$L(r, x_t) = \sum_t \left\| I_t - \sum_i \sum_j (r_i x_{ijt}) *^T P_{\ell_2, 1.0} (G_j U_i) \right\|_2^2 + \gamma |r|_1 + \lambda_G \sum_j \|G_j\|_2^2 + \lambda_U \sum_i \|U_i\|_2^2 \tag{4}$$

with $r, x \geq 0$. The first term is the mean-squared reconstruction error. The other terms include a sparsity penalty on $r$ and weight decay for $G$ and $U$. To stabilize learning we project each $B_{ij} = G_j U_i$ to unit $\ell_2$ norm ($P_{\ell_2, 1.0}$).

Inference for BSC is typically performed by initializing $x$ to some canonical vector and then alternatively optimizing $r$ and $x$ [3]. One of the issues with this approach is that the canonical vector might be a poor approximation to the true underlying pose parameters, especially in the case of independent features as in our model. We convolve each feature $B_{ij}$ with the image to produce a feature map $\alpha_{ijt} = B_{ij} * I_t$. We then project onto some appropriately chosen norm ball to compute $x_{ijt} = P_{\ell, \rho} (\alpha_{ijt})$.[2] Inference proceeds by alternatively optimizing $r$ and $x$ until convergence. To optimize $r$, we use iterative thresholding, while $x$ is optimized by projected gradient descent. Both sets of coefficients are forced to be non-negative, using a rectifier for $r$ and projecting on the positive part of the norm ball for $x$.

## 3  Experiments

For our experiments, we used $1920 \times 1080$ resolution YouTube videos converted to gray scale and scaled down to $236 \times 176$ pixels per frame. The frames were normalized using subtractive normalization[3]. We extracted sequences of 5 consecutive frames, with $r$ assumed to be constant for each sequence during training. We excluded sequences in the largest $5\%$ of Euclidean norm difference between frames to exclude sudden camera changes or changes between scenes. We used a stride of half the size of the kernel for transposed convolutions.

Our model learns localized oriented Gabor-like features similar to those seen in sparse coding. Figure 1 shows a subset of the learned $12 \times 12$ pixel features: each column shows $B_{i:}$ corresponding to different transformed versions of the same underlying feature. Note that not only translations but other transformations are learned as well, e.g., rotations and warping. The learned bilinear features allow accurate reconstruction, as seen for an example input in Figures 2(a) and 2(b). All feature sets were $2\times$ overcomplete.

To test whether each $B_i(\mathbf{x}_i)$ corresponds to a "steerable" filter, we visualize in Figures 3(a-e) a subset of the different instantiations (with different $\mathbf{x}_i$'s) of each feature across different inputs and image locations from our natural videos. Note that the model captures a wide range of such instantiations. To determine whether each $B_i(\mathbf{x}_i)$ captures the progression of a single underlying feature across frames, we visualized the evolution of features across sequences of frames. As seen in Figure 4, the learned features evolve across frames in a manner similar to spatiotemporal filters in the visual cortex, e.g., direction-selective Gabor filters moving in a particular direction.

## 4  Conclusion & Future Work

We extend the bilinear sparse coding model to handle large images and independent feature transformations. Our model learns to group similar features together, leveraging the smoothness of natural videos. Perhaps the most interesting direction for future work is that of extending this approach hierarchically.

---

[2]This allows us to use the features themselves to derive a suitable pose vector. For the projection of $x$ we use the simplex $S_\rho$ : $\sum_j |x_j| \leq \rho, |x_j| \geq 0$; the radius $\rho$ determines how sparse the coefficients will be.

[3]A Gaussian kernel is used to estimate the mean intensity around each pixel, which is then subtracted from the pixel value.

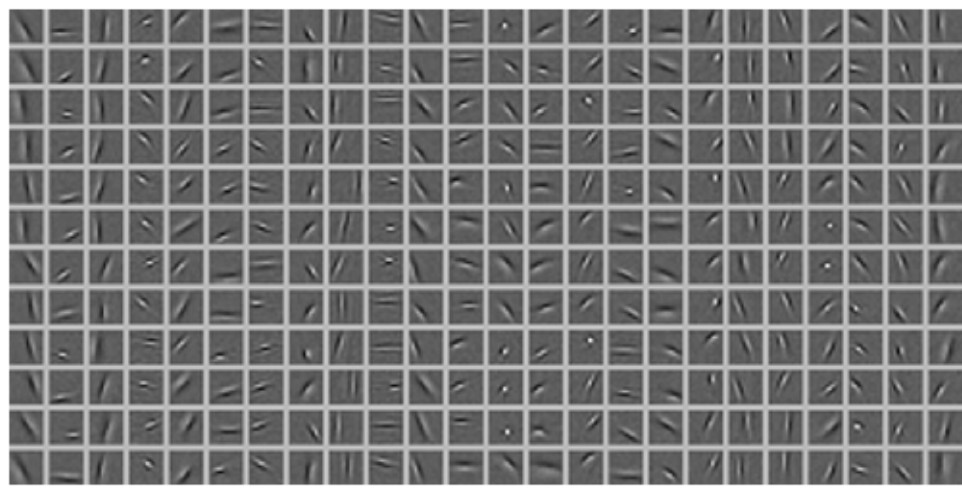

*Figure 1:* **Independent Bilinear Sparse Coding for Natural Videos**. $12 \times 12$ *pixel features* $B_{ij}$*: each column shows a feature* $i$ *for different* $j$*'s.*

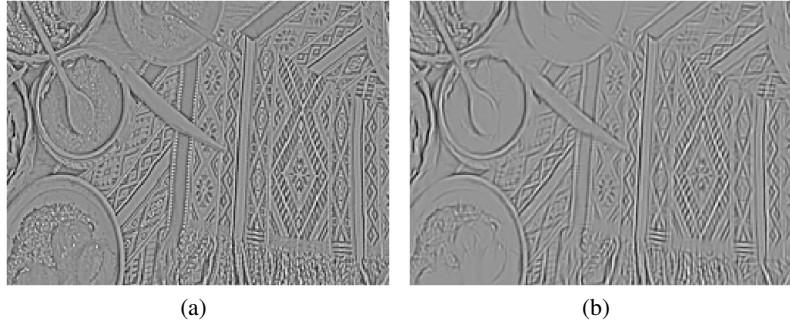

(a)                                         (b)

*Figure 2:* **Example Frame Reconstruction**. *(a) Original image and (b) its reconstruction using the learned bilinear features.*

## 5   Acknowledgments

This work was supported by NSF grant no. EEC-1028725, CRCNS/NIMH grant no. 1R01MH112166-01, and a grant from the Templeton World Charity Foundation (TWCF).

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
