# OpenReview forum: "Learning a Convolutional Bilinear Sparse Code for Natural Videos"
_NeurIPS.cc/2019/Workshop/Neuro_AI — Real Neurons & Hidden Units @ NeurIPS 2019 Poster_

### Official Review · AnonReviewer2 · 2019-09-26
**Interesting connection between bilinear models and neuro but no quantitative analysis of model.**

**Clarity:** 3

**Comment:**

The main place to improve is to have some quantitative analysis of the quality of their model perhaps MSE of image reconstruction. Then this evaluation could be used to study impacts of the parameters of their model which could then lead to neural hypotheses.

They have some qualitative evaluation in images of filters but they could explore the parameter space to understand what led to these features.

One of their stated novel contribution was that their filters were convolutional but they do not discuss the potential connection convolutional filters have to transformation of features which seemed like a gap. Weight sharing across shifted filters separates out feature and position yet many of their learned transformations are also translations. Is this an issue of spatial scale? This warranted some potentially interesting discussion though admittedly 4 pages isn’t a lot of space.

**Category:**

Neuro->AI

**Clarity Comment:**

Paper was organized, figures clear and readable. Some development of the model could have been left to the references and didn't add much to their contribution (e.g. Taylor approximation to a Lie model) .

When they say ‘steerable’ filter I was a little confused, do they just mean the basis vectors learned vary smoothly with respect to some affine transform parameter?

Their statement of the novelty of their method: ‘(1) allowing each feature to have its own transformation’ was not clear. Does this mean previous methods learned the same transformation for all features.


**Evaluation:**

2: Poor

**Importance:**

2: Marginally important

**Importance Comment:**

They make modifications to an existing generative model of natural images. They do not make direct comparisons to previous models or study quantitatively the results of the model with respect to its parameters. It is difficult to judge whether the new model is important because it has not been evaluated except by eye it does seem to reconstruct an image.

**Intersection:**

3: Medium

**Intersection Comment:**

They make an interesting connection to speed of processing that rapid changes better represented by the magnocellular pathway would be associated with transformations and slow parvo with identity. It was not clear though where they experimentally varied/tested this prior in their algorithm. So while an interesting connection they did not make clear where they substantively pursue it.

They draw an analogy between the ventral and dorsal stream of cortex and bilinear models of images.

**Rigor Comment:**

They show images of a single reconstruction but no quantification of reconstruction quality or comparison to previous methods. In the spirit of insight it would have been very nice to have a quantification of error with respect to parameters (priors on slow identity, fast form).  If it had been evaluated and its efficacy varied in an interesting way with respect to the parameters of the model this could be a potentially important model to understand why the nervous system trades off between object identity associated features, transformation features, and speed.

The statement that: ‘GAN’s and VAE features are not typically interpretable.’ Seemed broad and was unsupported by any citations and to my knowledge GAN’s and VAE’s have been used specifically to find interpretable features.




**Technical Rigor:**

1: Not convincing

---

### Official Review · AnonReviewer1 · 2019-09-26
**Bilinear sparse coding model with dynamic features, but unclear if dynamics work**

**Clarity:** 3

**Category:**

Common question to both AI & Neuro

**Clarity Comment:**

Figure legends/descriptions are too short, not totally clear what is shown in Figure 2.

**Evaluation:**

2: Poor

**Importance:**

2: Marginally important

**Importance Comment:**

Interesting extension to bilinear sparse coding models, but there is insufficient evidence in the work to support the claims in the abstracts - particularly that it captures the statistics of the transformations between frames.

**Intersection:**

3: Medium

**Intersection Comment:**

Unsupervised approaches might be interesting to some in the AI community.

**Rigor Comment:**

There are no quantifications of the performance of the model particularly in comparison to the original model that they are extending. The reconstruction of a single image in Fig1e is not a convincing test of the model - one would want to see how well the feature dynamics predict the next frame, if they are indeed sufficient to capture changes in the videos frame by frame.

**Technical Rigor:**

2: Marginally convincing

---

### Official Review · AnonReviewer3 · 2019-09-27

**Clarity:** 4

**Comment:**

This was a sensible algorithm for unsupervised feature learning, algorithm and results were clear, and results were reasonably good.


**Category:**

AI->Neuro

**Clarity Comment:**

The writing was very good, and the algorithm and results were very clearly presented, especially considering length constraints.


**Evaluation:**

4: Very good

**Importance:**

3: Important

**Importance Comment:**

This paper continues a line of work from the 2000s that has not had significant recent interest. I am glad it is getting tried with modern compute scale and tools, and I believe the results are promising. This submission is not sufficient on its own to convince me though that this approach will tell us new things about the brain or about artificial neural networks.

**Intersection:**

4: High

**Intersection Comment:**

The paper presented an unsupervised machine learning algorithm, which was used to try to describe representation learning in the brain.


**Rigor Comment:**

The algorithm was presented very clearly, and I believe all claims to be correct. I was surprised that x was set by projection rather than inference, and would have liked better understanding for why this was effective or desirable (though this may not be possible w/in length constraints).

**Technical Rigor:**

4: Very convincing

---

### Decision · Program_Chairs · 2019-10-02

Accept (Poster)